# Complementary Effects of Carbamylated and Citrullinated LL37 in Autoimmunity and Inflammation in Systemic Lupus Erythematosus

**DOI:** 10.3390/ijms22041650

**Published:** 2021-02-06

**Authors:** Roberto Lande, Immacolata Pietraforte, Anna Mennella, Raffaella Palazzo, Francesca Romana Spinelli, Konstantinos Giannakakis, Francesca Spadaro, Mario Falchi, Valeria Riccieri, Katia Stefanantoni, Curdin Conrad, Cristiano Alessandri, Fabrizio Conti, Loredana Frasca

**Affiliations:** 1Pharmacological Research and Experimental Therapy Unit, National Centre for Pre-Clinical and Clinical Drug Research and Evaluation, Istituto Superiore di Sanità, 00161 Rome, Italy; roberto.lande@iss.it (R.L.); anna.mennella@guest.iss.it (A.M.); raffaella.palazzo@iss.it (R.P.); 2Department of Oncology and Molecular Medicine, Istituto Superiore di Sanità, 00161 Rome, Italy; immacolata.pietraforte@iss.it; 3Department of Dermatology, University Hospital CHUV, 1011 Lausanne, Switzerland; curdin.conrad@chuv.ch; 4Department of Clinical and Internistic Sciences, Anesthesiology and Cardiovascular Sciences, University Sapienza, 00161 Rome, Italy; francescaromana.spinelli@uniroma1.it (F.R.S.); valeria.riccieri@uniroma1.it (V.R.); katia.stefanantoni81@gmail.com (K.S.); cristiano.alessandri@uniroma1.it (C.A.); fabrizio.conti@uniroma1.it (F.C.); 5Department of Pathology, University Sapienza, 00161 Rome, Italy; konstantinos.giannakakis@uniroma1.it; 6Confocal Microscopy UNIT, Core Facilities, Istituto Superiore di Sanità, 00161 Rome, Italy; francesca.spadaro@iss.it; 7National AIDS Center, Istituto Superiore di Sanità, 00161 Rome, Italy; mario.falchi@iss.it

**Keywords:** LL37, post-translational modifications (PTM), systemic lupus erythematosus (SLE)

## Abstract

LL37 acts as T-cell/B-cell autoantigen in Systemic lupus erythematosus (SLE) and psoriatic disease. Moreover, when bound to “self” nucleic acids, LL37 acts as “danger signal,” leading to type I interferon (IFN-I)/pro-inflammatory factors production. T-cell epitopes derived from citrullinated-LL37 act as better antigens than unmodified LL37 epitopes in SLE, at least in selected HLA-backgrounds, included the SLE-associated HLA-DRB1*1501/HLA-DRB5*0101 backgrounds. Remarkably, while “fully-citrullinated” LL37 acts as better T-cell-stimulator, it loses DNA-binding ability and the associated “adjuvant-like” properties. Since LL37 undergoes a further irreversible post-translational modification, carbamylation and antibodies to carbamylated self-proteins other than LL37 are present in SLE, here we addressed the involvement of carbamylated-LL37 in autoimmunity and inflammation in SLE. We detected carbamylated-LL37 in SLE-affected tissues. Most importantly, carbamylated-LL37-specific antibodies and CD4 T-cells circulate in SLE and both correlate with disease activity. In contrast to “fully citrullinated-LL37,” “fully carbamylated-LL37” maintains both innate and adaptive immune-cells’ stimulatory abilities: in complex with DNA, carbamylated-LL37 stimulates plasmacytoid dendritic cell IFN-α production and B-cell maturation into plasma cells. Thus, we report a further example of how different post-translational modifications of a self-antigen exert complementary effects that sustain autoimmunity and inflammation, respectively. These data also show that T/B-cell responses to carbamylated-LL37 represent novel SLE disease biomarkers.

## 1. Introduction

The activation of autoreactive T- and B-cells characterizes Systemic Lupus Erythematosus (SLE), a rare but potentially severe disease, with poor therapeutic options [1]. In this context, a deeper knowledge of the molecular pathways leading to autoimmunity in SLE may shed light over disease aspects still neglected and uncover novel disease biomarkers and therapy targets. The human cathelicidin antimicrobial peptide, (AMP, Uniprot P49913)—also called CAP-18 or FALL-39—is encoded by the human gene CAMP. The mature form, peptide LL37, which corresponds to the COOH-terminal part of the molecule (res. 134–170), exerts antimicrobial activity and acts as an immune modulator [2,3]. LL37 also acts as a T- and B-cell autoantigen in Systemic Lupus Erythematosus (SLE), as well as in psoriasis and the associated psoriatic arthritis (PsA) [4,5,6]. We previously hypothesized that LL37 easily acts as an autoantigen in psoriasis, PsA and SLE for three main reasons. The first, because LL37 is continuously and aberrantly expressed in lesional psoriasis skin, PsA joints and SLE affected skin and kidney, due to massive release by neutrophils or over-production by keratinocytes (especially in psoriasis) [4,5,6,7]. The second, because of the presence, in the LL37 sequence, of suitable “binding motifs” for the most common HLA-Class I and HLA-class II molecules, included the psoriasis associated Cw6*02 and the SLE associated HLA-DR15/DRB5 alleles [4,6]. This implies that anti-LL37 epitopes can be easily presented to T-cells in various HLA-contexts. The third, because LL37 plays a role as “danger” signal in complex with self-nucleic acids, creating a milieu that alters regulatory mechanisms and predisposes to autoreactivity [4,6,8,9,10,11]. An additional pathogenic feature, which can favor autoimmunity, is that LL37 presents various arginines and lysines in its sequence, amino acid residues acting as substrates for two irreversible post-translational modifications (PTM), citrullination and carbamylation, taking place during inflammatory processes [12,13,14]. In the citrullination and carbamylation processes, arginines and lysines/leucines are substituted by citrullines and homocitrullines, respectively. These PTM occur in inflammatory milieu dominated by neutrophil infiltration and/or neutrophil-extracellular traps (NET)/NET-like release [12,13,14]. Both PTM are permanent and change the functions of the original molecule [12,13,14]. We recently uncovered that citrullinated-LL37 (cit-LL37) acts as a more efficient antigen in SLE than native LL37 and likely favors establishment of autoreactivity to the original unmodified antigen, a phenomenon described for different antigens and PTM in other autoimmune settings [6,15,16]. Thus, we believe that, at least in some genetic HLA-backgrounds, citrullination of autoantigens may concur to the rupture of tolerance to self-antigens in SLE [6]. Carbamylation has similarity with citrullination. Following the citrullination process, the amino acid arginines are converted to citrullines by the peptidylarginine deiminases (PADs) enzymes, which may be released by activated neutrophils [17]. Carbamylation consists in the leucines and lysines substitutions with homocitrullines and it is a non-enzymatic reaction, which is still dependent on the neutrophil-derived enzyme myeloperoxidase (MPO) [18]. Notably, circulating antibodies to carbamylated proteins other than LL37 have been detected in SLE, although they have been traditionally studied in rheumatoid arthritis (RA), a disease in which this type of antibody reactivity is frequently detected [19,20,21,22].

In the present study we have addressed the T/B-cell immunogenicity, as well as the innate immune cell stimulatory effects, of what we call a “heavily carbamylated-LL37” in complex with DNA, with the aim to clarify whether carb-LL37 works side-by-side to cit-LL37 to favor autoimmunity and inflammation in SLE. The data reveal that carb-LL37 can be generated in SLE tissues and anti-carb-LL37 antibody reactivity is detectable in SLE sera and correlates with disease activity, measured as systemic lupus erythematosus diseases activity index (SLEDAI) [23]. Carb-LL37 stimulates T-cells with a specificity at least partially directed to the native antigen. Unlike cit-LL37, carb-LL37 is less stimulatory for T-cells but is still able to exert its activatory effect on plasmacytoid dendritic cells (pDCs) and B-cells when in complex with self-DNA, even when heavily carbamylated [13].

## 2. Results

### 2.1. Carb-LL37 and Anti-Carb-LL37 Antibodies Are Detectable in SLE

We have demonstrated that LL37 represents a frequently recognized autoantigen in SLE and that anti-LL37 antibodies correlate with SLEDAI [11]. More recently, we have repeated these observations and shown that anti-cit-LL37 antibodies are also frequent in SLE and correlate with the SLEDAI [6]. Carbamylation is a non-enzymatic PTM very similar to citrullination, which depends on cyanate reacting with any accessible primary amine, including that of the lysine side-chain and the leucine at the N-terminus of the LL37 peptide [13,18]. The reaction is stimulated by the enzyme myeloperoxidase (MPO), abundantly released by activated neutrophils. MPO converts thiocyanate into (iso)cyanate in the presence of hydrogen peroxide activity [18]. Given that the neutrophilic inflammation is characteristic of SLE [6,11,24], we made the hypothesis that carb-LL37 is generated in vivo. Indeed, when we searched for MPO and LL37 expression in SLE-target tissues (Appendix A), we found a consistent expression of both proteins, partly co-localizing in SLE kidney biopsies. We have demonstrated that both native LL37 and cit-LL37 can be present in SLE-affected tissues [6]. Here we stained SLE kidney biopsies with an antibody (RRB640) specific for carb-LL37 and not reacting to native LL37 or cit-LL37, as demonstrated in Appendix A. Figure 1a shows that carb-LL37 is indeed present in SLE-affected kidney biopsies and partially co-localizes with immunoglobulin (Ig)-G in these tissues, suggesting that SLE antibodies deposited in tissues could react to carb-LL37.

Given that SLE anti-carbamylated protein antibodies were shown to correlate with SLEDAI [21], we determined whether anti-carb-LL37 antibody reactivity was present in SLE sera. To address this, we used a carb-LL37 and a control carbamylated reverse (REV) LL37 (carb-REV), as antigens, in enzyme-linked immunosorbent assays (ELISA). Carb-LL37 was carbamylated at several sites [13]; (see Methods for sequences of antigens and control antigens used in this study). Figure 1b shows significant SLE serum antibody reactivity to carb-LL37, as measured by our in-house ELISA test [6,11], as compared to control healthy donors’ (HD) serum reactivity (See Appendix A for the characteristics of the SLE cohort and HD). Of note, the antibody response to the control carb-REV peptide was also significant, as compared to response of HD but significantly lower than the response to carb-LL37. Noteworthy, twenty-eight out of 45 SLE patients (62%) showed an antibody response to carb-LL37, whereas 8 out of 47 (17%) SLE patients showed response to the control antigen carb-REV. Given the detection of carb-LL37 co-localizing with IgG in renal biopsies, we also addressed whether SLE patients with renal damage could differ from SLE with no sign or renal damage for anti-carb-LL37 responses (Appendix A). The results show that the patients with renal damage had significantly higher anti-carb-LL37 antibody reactivity (and non-significant anti-cit- and anti-native-LL37 antibody increase). As expected, the patients with renal damage were those showing significantly higher SLEDAI. Altogether, these results suggest the existence of a characteristic milieu, which can also be the kidney, where carbamylation of self-antigens, included LL37, could occur, supported by the significant antibody reactivity to carb-LL37 detected in circulation of the SLE patients analyzed.

### 2.2. Anti-Carb-ll37 Antibodies Correlate with Disease Activity

We next addressed whether anti-carb-LL37 antibodies could mark active/severe SLE and therefore we correlated the magnitude of the anti-carb-LL37 antibody reactivity to the SLEDAI. A significant correlation was appreciated (Figure 2a). As control, we tested antibody reactivity to the native LL37 and cit-LL37 in the same SLE patients and we confirmed that both responses also correlated with SLEDAI, as expected [6,11] (Figure 2b). Interestingly, antibody reactivity to carb-LL37 correlated significantly with anti-cit-LL37 antibody reactivity (Figure 2c) and less but still in a significant manner, with anti-native LL37 antibody reactivity Figure 2d, left panel. For comparison, we also show correlation between anti-cit-LL37 and anti-native LL37 antibodies (Figure 2d, right panel). These results suggest that anti-carb-LL37 antibody reactivity can be taken as SLE disease marker (similar to anti-native LL37 and anti-cit-LL37 antibody reactivity [6]). The highest correlation between anti-carb-LL37 and anti-cit-LL37 antibody reactivity may suggest that anti-carb-LL37 antibodies are, at least in part, cross-reactive to cit-LL37, as shown in other settings [25].

### 2.3. SLE T-Cells Proliferate to Carb-LL37 and Can Be Follicular T-Helper-Like (T_FH_) Cells

Subsequently, we tested whether carb-LL37 was able to stimulate SLE T-cells as cit-LL37 and native LL37 did [6]. We preventively verified that carb-LL37 could bind HLA-molecules and we choose to test binding to HLA-DR1 (DRB1*0101), as an example of the most diffuse HLA-DR alleles in Caucasians and HLA-DR15 (DRB1*1501), which is associated with SLE [26,27]. Binding of carb-LL37 to these alleles was equivalent to the binding efficiency of cit-LL37 (binding of native LL37 is also reported for comparison in Appendix A and Appendix A). BrdU-incorporation assays (see methods), revealed that carb-LL37 could induce a significant proliferation of SLE CD4 T-cells, although less efficiently than cit-LL37 [6] (Figure 3a). Eight out of 25 SLE patients (32%), responded to carb-LL37, as compared to 17 out of 31 (55%) and 16 out of 31 (52%), who responded to native LL37 and cit-LL37, respectively. T-cell responses to carb-LL37 and native-LL37 correlated moderately, Figure 3b, which suggests possible, although limited, T-cell cross-reactivity between carb-LL37 and native LL37. T-cell responses to cit-LL37 and carb-LL37 tended to correlate moderately, although not significantly, in this cohort (Spearman r = 0.41, P = 0.08, N = 22). Carb-LL37 specific T-cells showed an up-regulation of the T_FH_ cell marker CXCR5 [6,28], represented as fold-increase with respect to T-cells proliferating to the control carb-REV (Figure 3c). The gating strategy for BrdU-assays and concomitant measurement of percent of CXCR5 expression, in the proliferating T-cells is depicted in Appendix A. We found a positive and significance correlation between the magnitude of T-cell proliferation to carb-LL37 and up-regulation of the CXCR5 marker (Figure 3d). A correlation was also appreciated between response of CD4 T-cells to native and cit-LL37 and CXCR5 up-regulation by the proliferating cells (SI proliferation to native LL37 vs. CXCR5 fold increase: Spearman r = 0.82, P = 0.0001, N = 18; SI proliferation to cit-LL37 vs. CXCR5 fold increase: Spearman r = 0.5, P = 0.018, N = 18). Thus, these data suggest that carb-LL37 is immunogenic for SLE T-cells and that carb-LL37 proliferating T-cells could, at least in part, belong to the T_FH_ cell-compartment, which drives antibody and autoantibody production [6,28].

### 2.4. Carb-LL37-Driven T-Cell Proliferation Correlates with SLEDAI and with Autoantibodies to the Native LL37

To have an indication of whether anti-carb-LL37 specific T-cells could concur to disease, for instance by helping production of autoantibodies to LL37, we assessed whether the magnitude of the T-cell proliferation to carb-LL37 correlated with the SLEDAI, as well as with anti-native LL37 antibody reactivity. Worth of note, we previously demonstrated that anti-LL37 antibodies exert pathogenic functions [6,11]. We detected a moderate but significant correlation between the magnitude of the carb-LL37 directed responses with the SLEDAI or the autoantibodies to LL37 (Figure 4a,b). We report, as control, the correlation between cit-LL37 and native LL37 specific T-cell responses with the SLEDAI (Figure 4c) and with the anti-native LL37 antibody reactivity as well (Figure 4c).

Of note, the fold-increase of CXCR5 expression in SLE CD4 T-cells stimulated with carb-LL37 not only significantly correlated with the magnitude of antibody reactivity to anti-carb-LL37 (Spearman r = 0.41, P = 0.049, N = 17) but also with the anti-native LL37 antibody response (Figure 4e). In Figure 4f, we report, for comparison, the correlation between the CXCR5 fold-increase in T-cells stimulated with LL37 or cit-LL37 and the anti-LL37 autoantibody responses. Gating strategy for CXCR5 detection in carb-LL37-stimulated T-cells, by flow cytometry, is shown in Appendix A.

Altogether, the data suggest that carb-LL37 stimulates T-cells that may concur to autoantibodies production and possibly to disease pathogenesis.

### 2.5. Culture of SLE PBMC with Carb-LL37 Has the Potential to Activate T-Cells Specific for the Native LL37

We have previously demonstrated that culture of SLE Peripheral Blood Mononuclear Cells (PBMC) with cit-LL37 was able to expand T-cells specific for both the cit-LL37 and the native LL37 epitopes [6]. We demonstrated this by taking advantage of peptide-MHC-tetramer staining, using tetramers of HLA-DR1, HLA-DR15 or HLA-DRB5 molecules loaded with epitopes derived from the native LL37 sequence [6]. We have performed here similar experiments in SLE patients expressing the HLA-DR15 (HLA-DRB1*1501) and/or the DR16 alleles. Since both HLA-DR15- and HLA-DR16-positive individuals co-express DRB5 [26,27], we also used peptide MHC-tetramers of the HLA-DRB5*0101 molecules, loaded with a native LL37 epitope (see Methods for specific peptide-MHC-tetramers used in this study). When we stimulated SLE PBMC with carb-LL37, we detected expansion of T-cells specific for the native LL37 epitope. Indeed, activated CD38^high^CD4 T-cells could be stained by peptide-MHC-tetramers of DRB1*1501 molecules, loaded with the native LL37-P1 epitope (see epitope sequence in Methods’ section), as in Figure 5a and in cumulative data in Figure 5b; (the lower panel shows the expansion of the same T-cells by the native antigen). Some experiments were also performed using peptide-MHC-tetramers of HLA-DRB5*0101 molecules loaded with the same native LL37 epitope P1 and cumulative data are reported in Figure 5c. The gating strategy relative to these experiments is depicted in Appendix A.

The expansion of the native LL37-P1-specific T-cells was consistent, as specific peptide-MHC-tetramer staining was significantly increased with respect to the staining with a peptide-MHC-tetramer of the HLA-DR15 or HLA-DR5 molecules loaded with a control peptide antigen (Figure 5a–c). Interestingly, CD4 T-cells stained with the native LL37 peptide-MHC-tetramers show an increased expression of CXCR5, as compared to the CD4-T-cell population stained by the control peptide-MHC-tetramers (Figure 5d). We report the effect of cit-LL37 stimulation for comparison. The gating strategy for CXCR5 expression on LL37-specific T-cells stained with native LL37 peptide-MHC-tetramers is shown in Appendix A. Altogether, these findings suggest that carb-LL37 concurs to the activation of the autoreactive LL37-specific T-cells in SLE, showing a variable but significant expression of the T_FH_ marker CXCR5.

### 2.6. Carb-LL37, Unlike cit-LL37, Still Binds the DNA and Activates Immune Cells

LL37 is able to activate innate immune cells and B-cells after forming complexes with nucleic acids in a special molecular conformation, suitable for TLR stimulation [6,9,11,29]. PDCs, as well as B-cells, are both activated by LL37-DNA or LL37-RNA complexes [8,10,29]. LL37 modified by citrullination at all five arginines, was reported to loose ability to transforms the “cell-free” DNA into an inducer of IFN-I by human pDCs [14]. By using an electrophoretic mobility shift assay (EMSA), we found that carbamylation of LL37 does not completely abolish the LL37 DNA binding ability (Figure 6a), unlike full citrullination (Figure 6b). The reduction in the cationic charges due to carbamylation partly affect the DNA-binding ability (Appendix A), as the native LL37 binds to DNA at both 10 µM and 3 µM concentrations, whereas carb-LL37 binds at concentrations between 10 and 30 µM, losing the DNA binding ability at 3 μM (Figure 6a and Appendix A). Despite this, carb-LL37 in complex with DNA retained the pDC IFN-α stimulatory ability at concentrations between 10 and 30 µM (Figure 6c). As expected, cit-LL37 did not stimulate any IFN-I production by pDCs [14]. LL37-DNA complexes are also able to stimulate memory B-cells to mature into plasma cells [29]. When we performed a stimulation of purified human B-cells with both cit-LL37 and carb-LL37, as compared to native LL37, again, only carb-LL37 maintained LL37 ability to stimulate the B-cells to mature into plasma cells in our in vitro assay (Figure 6d), gating strategy in Appendix A. Thus, full carbamylation, unlike full citrullination of LL37, does not abolish the stimulatory effect of LL37-DNA complexes on pDCs and on B-cell differentiation. These results suggest that carb-LL37 can still favor triggering of DNA sensing receptors in cells of both the innate and adaptive immune system, namely DC and B-cells.

## 3. Discussions

In this study we provide the first evidence concerning the presence of carb-LL37 in SLE affected tissues and detected for the first time anti-carb-LL37 antibody and T-cell reactivity in SLE. The simultaneous identification of carb-LL37 and both MPO and native LL37 in SLE-affected tissues corroborates the idea that neutrophilic inflammation typical of SLE favors carbamylation pathways [18]. Interestingly, we find that the magnitude of both B- and T-cell responses to carb-LL37 correlate with the SLEDAI, which may indicate a role of anti-carb-LL37 directed responses as new disease biomarkers. Our findings reinforce the assumption that self-proteins carbamylation is an inflammatory pathway implicated in SLE pathogenesis [19,20,21,22].

Remarkably, carb-LL37-stimulated T-cells do express CXCR5, a T_FH_ cell marker [28] and correlate with the magnitude of the autoantibody response to the native LL37. This can be taken as an indication that carb-LL37 specific T-cells concur to the generation/maintenance of autoantibodies specific for LL37 in SLE. This is of relevance as LL37-specific autoantibodies were shown to exert several pathogenic functions: (a) they contribute to increase IFN-I and pro-inflammatory factor production by pDCs and myeloid DCs, respectively [6,8,9,10,11]; (b) they interfere with the degradation of NET/NET-like-structures in tissues, which is instead necessary to avoid persistence of self-antigens and cell-free DNA [11]; (c) like classic SLE Ig-immune-complexes, also anti-LL37-specific-Ig/LL37 complexes could drive disease after deposition in tissues, a phenomenon that causes inflammation and tissue damage in several organs [1]. The kidneys can be severely affected by this pathway and we have demonstrated presence of antibodies (IgG) co-localizing with both the native LL37 [6] and carb-LL37 in SLE-affected kidney. Autoantibodies can damage the organ in several ways, included activation of complement [30]. Thus, it is possible that carb-LL37, generated in inflamed SLE tissue, contributes to this Ig-immune complex deposition, acting as an antibody target. Of note, we have shown that complement C9 deposition occurs in SLE-affected kidney in our previous study [6].

Correlation data and peptide-MHC-tetramer experiments suggest that both antibodies and T-cells responding to carb-LL37 can be, at least in part, cross-reactive to the native LL37. Alternatively, these data may suggest a frequent concomitant activation of both the citrullination and the carbamylation pathways in SLE tissues, as a consequence of neutrophilic inflammation, with simultaneous production of both cit-LL37 and carb-LL37. In the present study LL37 represents a suitable model antigen to shed light on the role of such irreversible PTM of cationic molecules in SLE.

On the T-cells side, the finding that a seven-day culture of SLE PBMC in the presence of carb-LL37 was sufficient to detect SLE T-cells stained with peptide-MHC-tetramers loaded with a native LL37 peptide-epitope, suggests that production of carb-LL37 in an inflamed milieu may expand autoreactive T-cells. Carb-LL37 may also stimulate skin/kidney transmigrating T-cell reactivation in the SLE-affected tissues. These trafficking T-cells could contribute to the local inflammatory milieu by secreting pro-inflammatory cytokines, as an immune amplification pathway.

Of note, we have used LL37 citrullinated at all five arginine sites to mimic a milieu of high PAD enzymes activity [14]. High carbamylation of LL37 can be obtained by an in vitro reaction and consists in the substitutions of the first leucine, (at the NH_2_-end) and almost all lysines with homocitrullines [13]. To mimic full carbamylation, here we have used a synthetic carb-LL37, with such characteristics see Appendix A. Interestingly, the modified peptides still bind to HLA, at least to the diffuse HLA-DR1 and the SLE-associated HLA-DR15 alleles [6,26,27]. Carb-LL37 is a less efficient antigen compared to cit-LL37 in our hands, at least in the SLE-cohort analyzed. The reason for that is unclear. It is plausible that the position of the PTM can be important, as the substituted amino acids can be more or less in proximity of the HLA “anchor positions,” at putative T-cell receptor (TCR) contact sites: a charge reduction at the TCR contact sites may affect TCR reactivity/affinity [31,32]. Thus, the different stimulatory ability may depend on the “frame” in which the LL37/modified LL37 sequence binds to each restriction HLA-molecule and therefore to the specific TCR. Of course, distribution of HLA-alleles in the reference SLE population can affect these results. Data in the literature report that HLA-DRB1*1501, HLA-DRB1*0301, HLA-DRB5 and DRB1*1601 can be enriched in SLE populations [26,27]. We know from our previous studies that several LL37-derived T-cell epitopes bind to HLA-DRB1*1501 and DRB5 [6] or possess “anchor motifs” suitable for HLA-DRB1*0301 allele binding (also linked to SLE, inferred by predictions servers [33]).

Clearance of cell-debris and self-nucleic acids is of importance to avoid autoimmunity and scavenger pathways are defective in SLE [34,35]. For instance, there is evidence for a deficit of nuclease activity [34,35]. Self-nucleic acid persistence can hyper-activate the immune system, via endosomal TLRs, such as TLR7, TLR8 and TLR9 [36]. Of interest, although heavily carbamylated, LL37 still binds DNA and transforms cell-free DNA in a “danger” signal, with activation of pDCs, IFN-I release and B-cell activation leading to maturation in plasma-cells, the cells that can make high affinity antibodies [6,29]. In contrast, full LL37 citrullination not only abolishes the interferogenic potential of LL37 [14] but also abolishes the B-cell maturation into plasma cells. Thus, it is likely that carb-LL37 and cit-LL37 could complement each other effects on innate and adaptive immunity activation in SLE. If this also occurs in other autoimmune settings is under investigations. The intriguing possibility that distinct PTM of the same self-antigen differently impact the immune system but cooperate to favor autoimmunity has been suggested previously as a novel mechanism for immune-pathogenesis [37]. A study in a mouse model of arthritis has shown differential effects of carbamylated and citrullinated autoantigens, with a predominant role of carbamylation, in that case, for disease induction. Indeed, despite their functional distinctions, the peptides produced by the two different PTM were shown to complement each other, as carbamylation had an immune-activating effect, which enhanced the arthritogenic properties of the citrullinated peptides [37].

We admit that the reason why cit-LL37 and carb-LL37 acts differently with respect to DNA-binding ability is at the moment elusive. The carb-LL37, although heavily carbamylated, still possesses positive charges due to arginines (R) (as carbamylation occurs at lysines, K, residues). Vice versa, cit-LL37, although losing five R, substituted by citrullines, still presents K residues. Both peptides used in this study have a net charge of +1. So, the different DNA binding and stimulatory abilities are likely dictated by a different spatial conformation, as the K and R residues are differently distributed along the LL37-sequence. The exact molecular mechanisms, which differentiate the action of cit-LL37 and carb-LL37 functions with respect to DNA-binding and interferogenic abilities are currently under investigation.

Limitation of this study resides in the fact that T-cell reactivity and the antibody response have not been studied at the clonal level. Therefore, the cross-reactivity of the antibodies to carb-LL37 and cit-LL37 and that of the carb-LL37-specific T-cells with the native LL37 are only inferred by the results of the correlations analyses and from the ex vivo peptide-MHC-tetramer assays. We are aware that the pro-inflammatory role of carb-LL37-specific T-cells in SLE should be confirmed by examining the polarization profile of these cells. For the moment, we know that carb-LL37-specific T-cells express the T_FH_-like marker CXCR5 and that fold increase of this marker in carb-LL37 responding T-cells correlates with autoantibody production. The lack of HLA-typing of the whole SLE cohort can represent a weakness of the study. This issue can be addressed more systematically in a subsequent study. A further limitation is the lack of conclusive demonstration that carb-LL37-anti-carb-LL37-Ig-complexes circulate in SLE-blood. This is inferred by the staining in the tissues which may suggest deposition of these immune complexes in the kidney. As patients with kidney damage have more anti-carb-LL37 antibodies in plasma, it will be worth to systematically analyze, using matched kidney and blood sample pairs, whether anti-carb-LL37 antibodies can be biomarkers of kidney damage.

In conclusion, these results provide the first indication that carbamylation of LL37 can contribute to the generation of anti-native LL37 antibodies with potential pathogenic effects in SLE [6], via activation of T helper cells and a direct effect on B-cell differentiation. As a corollary, the correlation studies suggest that the adaptive immune responses to carb-LL37 may represent an additional SLE-disease marker [6]. This study is also the first to address the capacity of the two irreversible post-translational modification of LL37 (carbamylation and citrullination) to effect human B-cell stimulation and transition to plasma cells.

## 4. Materials and Methods

### 4.1. Human Studies and Samples

SLE blood (10 mL) and kidney biopsies were obtained from Policlinico Umberto I, Sapienza University, Rome, Italy. Sera from HD, matched for age/sex with the SLE patients, were from the Blood Center, Policlinico Umberto I, Italy. Disease activity in SLE patients was assessed by SLEDAI 2000 [23]. HLA-Class II typing (HLA-DR) of few SLE patients was performed in the reference center for HLA typing: Geneva University Hospital, Switzerland and “Centro Transfusionale,” Rome, Italy. We obtained all samples upon approval by the Ethic Committees of the Sapienza University. All blood and tissue donors gave informed consent, according to the Helsinki’s Declaration.

### 4.2. Antigens

LL37: (LLGDFFRKSKEKIGKEFKRIVQRIKDFLRNLVPRTES), was purchased from Proteogenix (Schiltigheim, Strasbourg, France).

REV LL37 (SETRPVLNRLFDKIRQVIRKEFEKGIKEKSKRFFDGL), cit-LL37, {LLGDFFR(cit)KSKEKIGKEFKR(cit)IVQR(cit)IKDFLR(cit)NLVPR(cit)TES}, cit-REV, {SETR(cit)PVLNR(cit)LFDKIR(cit)QVIR(cit)KEFEKGIKEKSKR(cit)FFDGLL}, carb-LL37 (L*LGDFFRK*SK*EK*IGKEFK*RIVQRIK*DFLRNLVPRTES} in which the asterisks indicate substitution with homocitrullines)

carb-REV: (SETRPVLNRLFDK*IRQVIRKEFEK*GIK*EK*SK*RFFDGLL*), in which the asterisks indicate substitution with homocitrullines, were all synthesized by Biomatik (Wilmington, DE, USA).

### 4.3. Isolation and Stimulation of Blood pDCs

Buffy-coats from HD were from Centro Trasfusionale, Policlinico Umberto I, Rome, IT. After separation of mononuclear cells by Ficoll centrifugation, pDCs were purified as described [11], by using the Diamond Plasmacytoid Dendritic Cell Isolation Kit (Miltenyi Biotec, Bergisch Gladbach, Germany) to obtain 99% purity. Purified pDCs were seeded into 96-well round-bottom plates at 3.5/5 × 10^4^ cells per well in 200 μL RPMI 1640 (GIBCO), supplemented with 10% fetal calf serum (FCS), 1% Hepes and 1% sodium pyruvate. LL37, cit-LL37 and carb-LL37 were premixed with total human genomic DNA (extracted from PBMC of HD, see below) (15 μg/mL). After 15 min incubation at room temperature the mix was added to the pDC cultures. IFN-α in supernatants was measured by ELISA (MabTech, Stockholm, Sweden), after 24 h (according to the manufacturer’s instructions).

### 4.4. ELISA for Autoantibody Detection in Sera and Culture Supernatants

Anti-native LL37/cit-LL37/carb-LL37 antibodies were measured as described [6,11]. Briefly, 96-well flat-bottom plates (Non-Binding surface polystyrene, Corning, Glendale, AZ USA) were coated with 2 μg/mL of native LL37, cit-LL37 or carb-LL37 in carbonate buffer (0.1 M NaHCHO3, pH 9) for 2 h (or overnight) and washed four times with PBS plus 0.1% Tween-20. This washing buffer was used for washing at all steps. The blocking buffer containing 2% bovine serum albumin (BSA, Sigma) in PBS was used for at least 1 h (or overnight) to saturate unspecific binding sites. After washing, sera were diluted at various concentrations (usually at 1:100 or 1:200) in PBS + 2% BSA followed by 1 h incubation with a horseradish peroxidase–conjugated (HRP) goat anti-human IgG (Sigma-Aldrich), diluted 1:10,000 in PBS. The color was developed for 5 min with 3,3′,5,5′-tetramethylbenzidine (TMB) substrate (Sigma-Aldrich). The reaction was stopped by adding 50 μL of 2 N H2SO4 and absorbance determined at 450 nm with a reference wavelength of 540 nm. Responses were considered positive when the OD measured exceeded the mean OD values measured in healthy donors’ plasma plus two standard deviations (+2SD), calculated on the HD group values, as described [6,11].

### 4.5. Peptide-MHC Class II Binding Assays

Peptide-MHC-binding assays was performed by ImmunAware (Horsholm, Denmark). Binding was measured using a Luminescent Oxygen Channeling Immunoassay (LOCI), marketed by Perkin Elmer (Hopkinton, MA, USA) as AlphaScreen [6,38]. This is a nonradioactive bead-based homogenous proximity assay, where the donor beads containing a photosensitizer compound, upon excitation with light at a 680 nm wavelength, converts ambient oxygen to energy-rich, short-lived singlet oxygen; this latter is transferred to acceptor beads, in close proximity only. The acceptor beads can respond to a single oxygen with a luminescence/fluorescence cascade, leading to an amplified signal in the 520–620 nm range. There is a tag binding to the donor beads: a biotin group engineered into the HLA β chain.

The acceptor beads are linked instead to a conformation-dependent HLA class II specific antibody (L243). Thus, only when a peptide binds well the HLA class II molecule, a corrected folded HLA class II molecule is formed and can be recognized by the L243 antibody. Biotinylated recombinant HLA class II molecules were diluted from an 8 M urea buffer into a folding buffer (PBS, 20% glycerol, Protease inhibitor mix). Different peptide concentrations, between 0.13 nM and 10,000 nM, were added into the assay. The tested peptides were titrated in 5-to-8 five-fold concentration, from 20 μM to 0.3 nM. The reaction mixtures were incubated for 48 h at 18 °C to allow peptide-HLA-complex formation to reach a steady state. After incubation, 15 μL of the folding mixture was transferred to a 384-well Optiplate (Perkin Elmer), followed by addition of 15 μL of a solution containing L243 acceptor beads and streptavidin donor beads (final concentration of beads was 5 μL/mL). The plates were incubated over night at 18 °C and then measured in enVisionTM, Perkin Elmer. Assay signals measured in OD450 were plotted against the offered peptide concentrations and analyzed by non-linear regression using GraphPad Prism. The peptide concentration resulting in half saturation, the half maximal effective concentration (EC50), was estimated by fitting the experimental data to the equation: Y = Bmax*X/(KD + X), where Y is the OD450 measurement of complexes formed and X is the concentration of ligand (peptide) offered. The EC50 approximates the KD as long as the receptor concentration used is less than the KD, thus avoiding ligand depletion.

### 4.6. Antibodies and Reagents for Flow Cytometry, Confocal Microscopy

Antibodies to CD4, CD8, CD3 conjugated with various fluorochromes (FITC, phycoerythrin (PE), peridinin-chlorophyll-protein (PerCp), allophycocyanin (APC), PE-Cy7) were from BD Biosciences or eBiosciences (San Diego, CA). For further LL37-specific T-cell characterization and B-cell characterization we used: APC-Cy7-CXCR5 (FAB190), APC or PE-CD38, CD19, CD20, CD27, antibodies, purchased from BD Biosciences, eBiosciences, Novus Biologicals (Littleton, CO), R&D (Minneapolis, MN). Appropriate isotype-matched controls were purchased from the same companies. PerCp-7-AAD was from BD Pharmingen, DAPI from Invitrogen (Thermo Fisher, Carlsbad, CA, USA).

The rabbit monoclonal antibody specific for carb-LL37 (clone RRB640) and the mouse antibody specific for native LL37 (Mab137), not cross-reacting to native/carb-LL37 and cit-LL37 were obtained, (and their fine specificity was tested) as previously described, in the Antibody Facility of UNIGE, Geneva, CH [6,39].

### 4.7. T-cells Proliferation Assay

PBMCs were purified from EDTA-treated blood, on Ficoll-Hypaque (Pharmacia Fine Chemicals, Uppsala, Sweden) and were incubated (10^5^ cells/well) in 96-well-flat-bottom-microplates (BD) in T-cell-medium (RPMI 1640, 10% heat-inactivated human serum, HS, Gibco), 2 mM L-glutamine, 10 U/mL penicillin and 100 μg/mL streptomycin), with/without peptides. We performed assays on fresh cells, within 1-to-3 h from collection or on PBMCs frozen in 90% FCS−10% DMSO, whose viability we assessed by Trypan blue exclusion, with an inverted microscope. Recovery of live cells was between 65 and 85% of the frozen number. At day 3 and 5, BrdU was added (1 μg/mL). BrdU-incorporation was detected by APC-labeled anti-BrdU antibody (BD Pharmingen), after surface staining for CD4, CD3, CD8 and analyzed by flow cytometry (Gallios, BeckmanCoulter, Brea, CA, USA). Internal control for T-cell viability and proliferation was PHA treatment (2 μg/mL). SI for proliferation was calculated by dividing percent of BrdU-staining in the presence of each peptide antigen (LL37, carb-LL37, cit-LL37, LL37 reverse, REV; cit-LL37 reverse, cit-REV; carb-LL37 reverse, carb-REV) by the percent obtained in the untreated cells. SI for proliferation was considered positive when the values of percent of BrdU positive cells was three times higher than the values obtained with the control peptides (SI > 3). Assays were repeated twice with each patient. In rare cases in which background proliferation (culture with REV, cit-REV, carb-REV), exceeded antigen-specific proliferation (that is response to LL37, carb-LL37, cit-LL37), the assay was repeated three times. Concomitant phenotype analysis of LL37-responder T-cells included staining for CD3, CD4, CD8, CXCR5 (CXCR5 was assessed on BrdU/CD3/CD4-positive cells).

### 4.8. Peptide-MHC-Tetramer Staining of Blood T-Cells

Peptide-MHC-tetramers were from ImmunAware (Denmark). Staining was performed for 40 min with 1microL of tetramer suspension at 37 °C, followed by staining for CD4 (4 °C, 20 min), without washing. After incubation with anti-CD4 antibody, cells were washed, resuspended in PBS and kept at 4 °C. Before flow cytometry-acquisition, cells were labeled with PercP-conjugated 7-AAD to exclude dead cells. The peptide-MHC-tetramers used in this study are: the HLA-DR15-P1: tetramer of HLA-DRB1*1501 LLGDFFRKSKEKIGK (corresponding to the first aa at the NH2-term of the mature native LL37), (PE-conjugated) and the control peptide-MHC-tetramer called DR15-ctr: tetramer of HLA-DRB1*1501 TNPLIRHENRMVLASTTM1-169-185 (PE-conjugated) [6]. The other couple of peptide-MHC-tetramers used were HLA-DRB5*0101 loaded with the same LL37 epitope as above and the control tetramer HLA-DRB5*0101 loaded with the control epitope PVSKMRMATPLLMQA (CLIP-derived) [6].

### 4.9. Laser Scanner Confocal Microscopy (LSM)

Paraffin-included SLE renal biopsies were obtained from Policlinico Umberto I, Rome, IT. Biopsies were de-paraffinized in xylene (5 min, two times), followed by passages in: absolute ethanol (3 min), 95% ethanol in water (3 min), 80% ethanol in water (3 min), 70% ethanol in water (3 min) and antigen retrieval (5 min at 95 °C in 10 mM sodium citrate, pH 6.0). After saturation with blocking buffer (PBS, 0.05% tween 20, 4% BSA) for 1 h at room temperature the biopsies were washed with washing buffer (PBS, 0.05% tween 20 and antibodies (Mab137 and RRB670) were added at 1:50 and 1:10 (10 µg/mL) dilution (all diluted in blocking buffer) (isotype control was added at 1:100 concentration), for 1 h at room temperature in a humidified chamber. Slides were washed three times for 3 min under agitation with washing buffer and incubated with donkey anti-mouse or anti-rabbit secondary antibody, conjugated with AlexaFluor 647, Alexa Fluor 568 or 594 or AlexaFluor 488 (Abcam, Cambridge, UK) depending on the combinations of markers analyzed (in humidified chamber). Slides were washed again in washing buffer and mounted in Prolong Gold anti-fade media containing a DNA dye (DAPI) (Molecular Probes), before analysis with a confocal microscope, objectives 20× or 40× and 60× in oil immersion. Images were taken by a FV1000 confocal microscope (Olympus, Tokyo, Japan), using a Olympus planapo objective 40× or 60× oil A.N. 1,42. Excitation light was obtained by a Laser Dapi 408 nm for DAPI, an Argon Ion Laser (488 nm) for FITC (Alexa 488), a Diode Laser HeNe (561 nm) for Alexa 568 and a Red Diode Laser (638 nm) for Alexa 647. DAPI emission was recorded from 415 to 485 nm, FITC emission was recorded from 495 to 550 nm, Alexa 568 from 583 to 628 nm and Alexa 647, from 634 to 750 nm. Images recorded had an optical thickness of 0.3 μm.

### 4.10. Production of Human DNA

Human DNA was extracted from PBMCs, as reported [40] and was fragmented by sonication. Five milligrams of DNA in a volume of 300 μL were fragmented using the Sonics Vibra Cell sonicator (Newtown, CT, USA), with the following settings: 2, 4 and 10 sonication cycles (30 s on, 30 s off, in ice) to obtain DNA fragment sizes between 100 and 1000 bp. The resulting size distribution was controlled by 2% agarose gel electrophoresis.

### 4.11. Binding of Peptides to DNA by Electrophoretic Mobility Shift (EMSA)

EMSA was performed by mixing various μM concentrations of LL37 or carb-LL37 or cit-LL37 (or AMP S100A8, as negative control [40]), with appropriate concentration of DNA, which respected the appropriate protein–DNA ratio used in pDC-stimulation assays. DNA alone and the mixtures were run on a 2% agarose gel to evidence the delay in migration of the DNA, due to the binding to the tested proteins. DNA on the gel was visualized by with SYBR Safe DNA gel staining (ThermoFisher Scientific).

### 4.12. Isolation of B-Cells from Buffy Coats

PBMCs were isolated by Ficoll-Paque centrifugation (as described above). Total B-cells were isolated by using the Human B-Cell Isolation Kit II, (Miltenyi Biotec, Bergisch Gladbach, Germany), from fresh PBMCs. The purity of B-cells was verified by flow cytometry using the monoclonal anti-human CD19 fluorescein isothiocyanate (FITC). The percentage of memory B-cells was assessed by flow cytometry using monoclonal anti-human CD19 fluorescein, anti-human CD27 allophycocyanin and anti-human CD38 Pacific Blue antibodies (BD Biosciences, San Jose, CA, USA). The majority of B-cells (>80%) were memory cells expressing CD19/CD27 and low CD38 [29].

### 4.13. Stimulation of Purified B-Cells

LL37/cit-LL37/carb-LL37 were each premixed with DNA (15 μg/mL) and added to the B-cell cultures in RMPI 10% FCS, after a 15-min incubation at room temperature. At day 2, hrIL-2 at 50UI/mL was added to the cultures, which were kept in the incubator for 7 days.

Maturation of B-cells into plasma cells was investigated after 7 days culture by flow cytometry (Gallios, BeckmanCoulter, Brea, CA, USA) following surface staining with the following monoclonal antibodies: anti-human CD19 fluorescein-, CD27 allophycocyanin-, CD38 Pacific Blue-conjugated (BD Biosciences, CA, USA). Analysis of the FCS files was performed by FlowJo (BD, San Jose, CA, USA) or by Kaluza (Beckman Coulter, Indianapolis, IN, USA).

### 4.14. Statistical Analyses

Differences between mean values were assessed by Wilcoxon’s matched-pair signed rank test, to compare responses to antigens and control antigens in the same donor. We used Mann-Whitney test for comparison of T-cell and antibody responses between groups of patients and HD. Statistical significance was set at *p* < 0.05. Correlation analyses were performed by Spearman’s rank-correlation test. Data were analyzed by GraphPad Prism 7.0 or SPSS software.

## Figures and Tables

**Figure 1 ijms-22-01650-f001:**
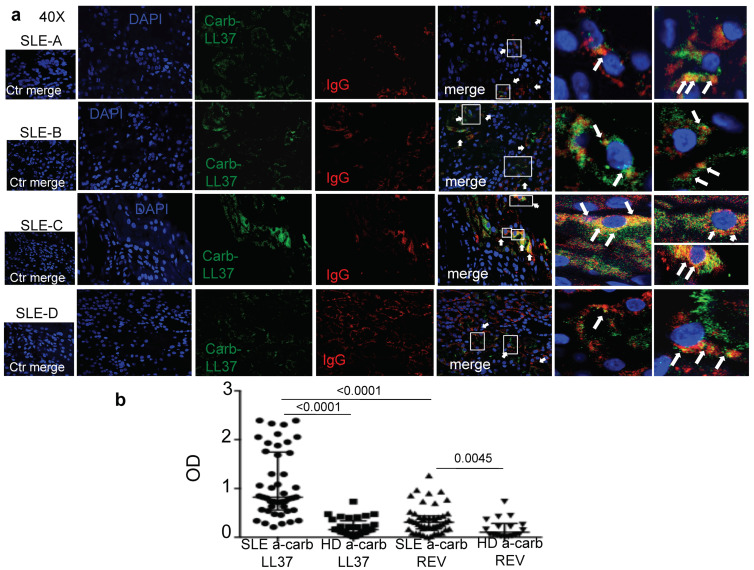
Carb-LL37 is present in Systemic Lupus Erythematosus (SLE) and is recognized by antibodies. (**a**) Representative confocal images of four out of six SLE-affected renal biopsies, showing expression of carb-LL37 and IgG. Two enlarged insets of confocal images for each SSc-patients, to evidence the co-localization between IgG staining and carb-LL37 staining are shown on the right. All stained biopsies were from lupus nephritis (class IV). White rectangoles represent the inset shown in the right side of the figure. White arrows show co-localization of carb-LL37 with IgG (**b**). Antibody reactivity of SLE and healthy donors’ (HD) sera to carb-LL37 and control carbamylated peptide REV (carb-REV), measured by enzyme-linked immunosorbent assays (ELISA) (and reported as optical density, OD). Medians with interquartile range are shown on the graph, *p*-values by Mann-Whitney’s test.

**Figure 2 ijms-22-01650-f002:**
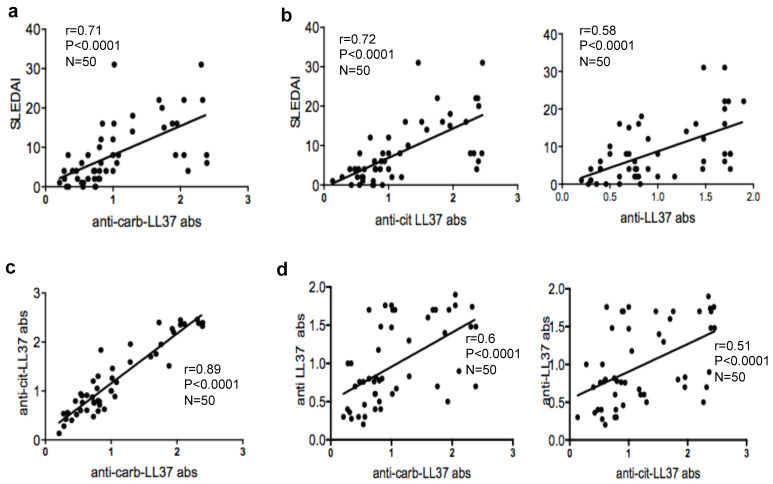
Anti-carb-LL37 antibodies correlate with disease activity. (**a**) Anti-carb-LL37 and (**b**) anti native LL37/anti-cit-LL37-antibody response (OD), (shown for comparison), plotted against systemic lupus erythematosus diseases activity index (SLEDAI) values. Spearman’s coefficient “r”, significance *p*, sample size N, indicated. (**c**) Anti-carb-LL37 and anti-native/cit-LL37 antibody responses, plotted against anti-cit-LL37 (**d**) or anti-carb/cit/native LL37-responses, as indicated. Spearman’s r, significance *p*, sample size N, indicated.

**Figure 3 ijms-22-01650-f003:**
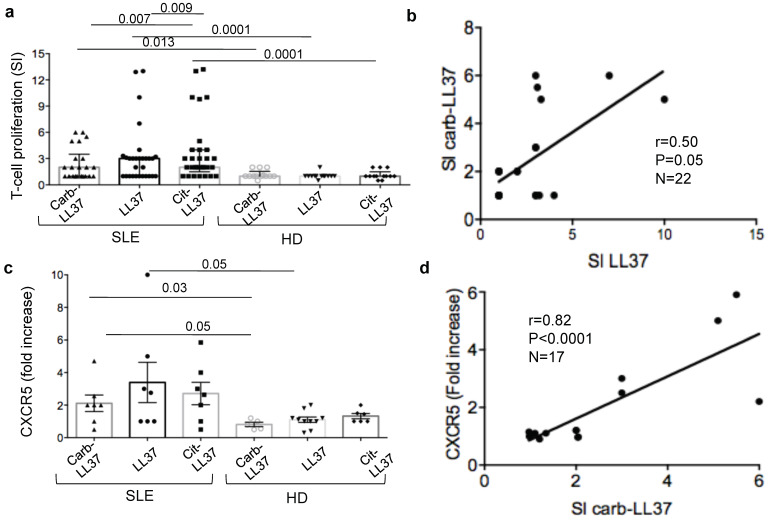
Carb-LL37 induces SLE T-cell proliferation. (**a**) Carb-, native and cit-LL37 induced SLE T-cell and HD T-cell proliferation, expressed as stimulation indexes (see Methods, SI). Horizontal bars represent the means, vertical bars standard errors of the mean, *p*-values by Wilcoxon’s test. SLE patients tested with carb-LL37 (N = 25), LL37 (N = 31), cit-LL37 (N = 31); HD = 14; (**b**) Proliferation (as SI) to carb-LL37 plotted against T-cell proliferation to native LL37 (SI LL37). Spearman’s r, significance *p*, sample size N, indicated. (**c**) CXCR5 expression (expressed as fold increase with respect to untreated T-cells) of BrdU^+^CD4^+^ T-cells, from SLE patients and HD, assessed by flow cytometry. Horizontal bars represent the means, vertical bars standard errors of the mean, *p*-values by Wilcoxon’s test. (**d**) CXCR5, expressed as fold increase with respect to expression in untreated cells, of CD4 T-cells from SLE patients or HD. Horizontal bars represent the means, vertical bars are standard errors of the mean, *p*-values by Wilcoxon’s test.

**Figure 4 ijms-22-01650-f004:**
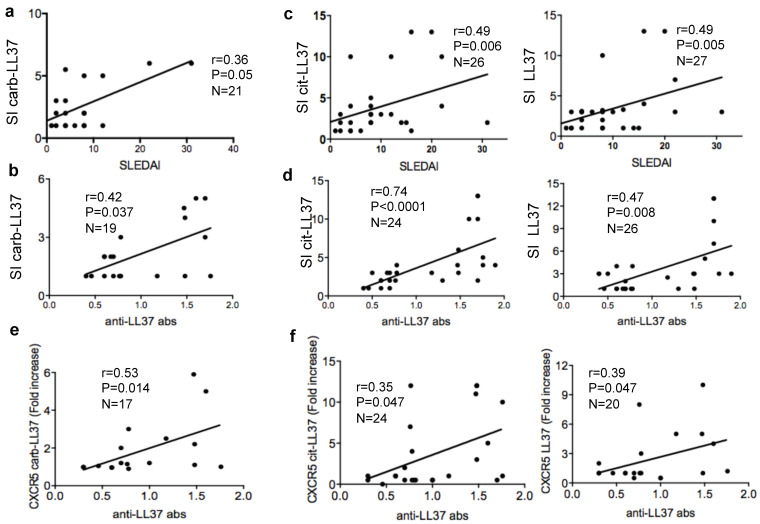
Carb-LL37 proliferating SLE T-cells correlate with SLEDAI and with anti-native LL37 autoantibodies. (**a**) Proliferation to carb-LL37 expressed and SI plotted against SLEDAI or (**b**) anti-native LL37 antibody reactivity (anti-LL37 bas). (**c**,**d**) Proliferation to cit-LL37 (left panel) or to native LL37 (right panel), expressed and SI, plotted against SLEDAI (**c**) or against anti-native LL37 antibody reactivity (**d**). (**e**,**f**) CXCR5 expression (as fold-increase with respect to untreated T-cells), in SLE T-cells, in response to carb-LL37 (**e**) or to either cit-LL37 or native LL37 (**f**), plotted against the anti-LL37 antibody reactivity (OD). In all panels Spearman’s r, significance *p*, sample size N, are indicated.

**Figure 5 ijms-22-01650-f005:**
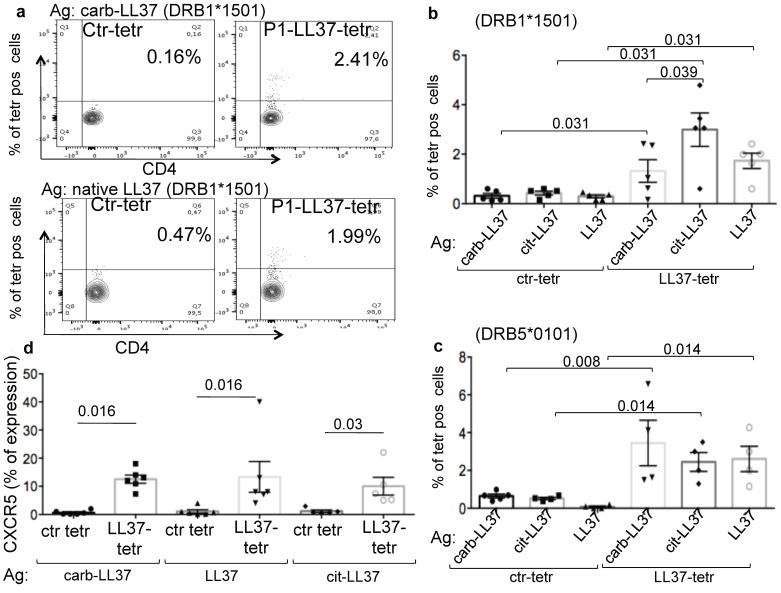
SLE-blood T-cell stimulation with carb-LL37 expands T-cell specific for native LL37. (**a**–**c**) Peripheral Blood Mononuclear Cells (PBMC) of SLE patients were stimulated by carb-LL37, cit-LL37 or native LL37 as indicated and, after 7 days, T-cells were stained with peptide-MHC-tetramers of HLA-DR15*1501 (**a**,**b**) or HLA-DRB5*0101 (**c**), loaded with a native LL37 epitope (P1) or a control peptide-MHC-tetramer (see Methods). Cells were analyzed by flow cytometry and percent of peptide-MHC-tetramer staining is reported on the contour plots in (**a**) or on the *y*-axis in (**b**,**c**) (cumulative data). (**d**) SLE T-cells were stimulated with carb-LL37, cit-LL37 or native LL37 and percent of expression of CXCR5 on peptide-MHC-tetramer-positive cells was assessed by flow cytometry. In the panels (**a**–**c**) horizontal bars represent the mean, vertical bars standard errors of the mean, *p*-values by Wilcoxon’s test.

**Figure 6 ijms-22-01650-f006:**
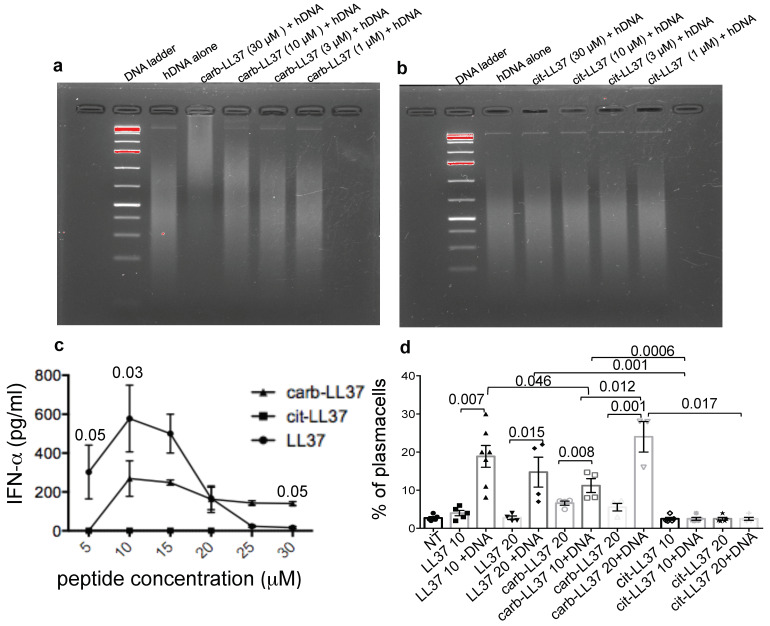
Carb-LL37, unlike cit-LL37, binds DNA and activates pDCs and B-cells. (**a**,**b**) EMSA, performed with carb-LL37 (**a**) and cit-LL37 (**b**), to show binding to DNA on a 2% agarose gel. Representative experiments of 3-to-5 performed. (**c**) Purified pDCs were stimulated with different μM concentrations of carb-LL37, cit-LL37 or native LL37 pre-complexed with human DNA (the peptides alone and DNA alone induced no IFN-α response, therefore they are not reported in the picture). (**d**) Carb-LL37-, cit-LL37- and native LL37-DNA complexes or each peptides alone, were used to stimulate the differentiation of memory B-cells into plasma cells over a 7 days culture (in the presence of 50 UI/mL of hrIL-2). Percent of plasma-cells (CD19^−^CD27^high^CD38^high^ cells) was evaluated by flow cytometry. Cumulative data relative to 4-to-7 different experiments. Horizontal bars are the means; vertical bars are standard error of the mean; *p*-values by Student’s t test for paired samples.

## Data Availability

Data relative to this work are available upon reasonable request to the corresponding author L.F.

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
