# Peer review of "Complementary Effects of Carbamylated and Citrullinated LL37 in Autoimmunity and Inflammation in Systemic Lupus Erythematosus"

_ijms, 2021, doi:10.3390/ijms22041650_

Round 1
Reviewer 1 Report
It is very interesting paper with new aspects in SLE molcular pathogenesis but in my opinion method sectiom has to be improved and be more clear. Also the number of renal biopsies is rather low.
Author Response
Please see attach

Reviewer 2 Report
In this manuscript, Lande and colleagues study the role of the posttranslational modification, carbamylation, in the immunogenicity of the cationic peptide LL37 that has been well-studied in autoimmune conditions such as SLE. Carbamylation is a PTM that has not been systematically studied in SLE. Authors document the presence of antibodies against carbamylated version of LL37, these antibodies correlate with disease activity although anti-citrullinated and naïve versions of LL37 also associate. The three versions of LL37 trigger T cell proliferation. Interestingly carLL37 binds DNA while cit LL37 doesn’t and this leads to the capacity of these modified LL37 to differentially triggers IFN production by pDCs.
This is a well-written manuscript and authors clearly state that possibility of cross-reactivity between carbamylation and citrullination, since this has been documented and also been the main reason for the limited studies among these posttranslational modifications in different conditions such as psoriasis, RA among others.
Minor Point:
- The co-localization of carLL37 and IgG provided in Fig 1 are not convincing, since most of the staining are NOT co-localizing (detecting carLL37 and IgG are against that particular version of LL37). Authors should be careful about the specificity of the antibody to recognize carLL37 in tissue by IF.
- In Fig6d, author should include 20 uM concentration for LL37 since authors are using 10 and 20 uM for car- and cit-LL37.
- Is there any evidence carLL37-IgG immune-complex formation in SLE? Is there any association of anti-CarLL37 to kidney damage?
- Why carLL37 still bind to DNA while cit-LL37 doesn’t when under both conditions positive charges are lost? This should be discussed.
Author Response
Please see attach
